# Discrepancies in Karst Soil Organic Carbon in Southwest China for Different Land Use Patterns: A Case Study of Guizhou Province

**DOI:** 10.3390/ijerph16214199

**Published:** 2019-10-30

**Authors:** Zhenming Zhang, Xianfei Huang, Yunchao Zhou, Jiachun Zhang, Xubo Zhang

**Affiliations:** 1Institute for Forest Resources & Environment of Guizhou, Key Laboratory of forest cultivation in plateau mountain area, College of Forestry, Guizhou University, Guiyang 550025, China; zhangzhenming1986@163.com (Z.Z.); hxfswjs@gznu.edu.cn (X.H.); 2Institute of biological research of Guizhou, Guizhou Academy of Sciences, Guiyang 550025, China; 3Guizhou Provincial Key Laboratory for Environment, Guizhou Normal University, Guiyang 550001, China; 4Guizhou Botanical Garden, Guizhou Academy of Sciences, Guiyang, Guizhou 550000, China; zhangjiachun1988@163.com; 5Institute of Geographic Sciences and Natural Resources Research, Chinese Academy of Sciences, Beijing 100101, China; zhangxb@igsnrr.ac.cn

**Keywords:** land uses, rock exposure, soil thickness, soil organic carbon, spatial distribution

## Abstract

The assessment of soil organic carbon (SOC) in mountainous karst areas is very challenging, due to the high spatial heterogeneity in SOC content and soil type. To study and assess the SOC storage in mountainous karst areas, a total of 22,786 soil samples were collected from 2,854 soil profiles in Guizhou Province in Southwest China. The SOC content in the soil samples was determined by the oxidation of potassium dichromate (K_2_Cr_2_O_7_), followed by titration with iron (II) sulfate (FeSO_4_). The SOC storage was assessed based on different land uses. The results suggested that the average SOC density in the top 1.00 m of soil associated with different land uses decreased in the following order: Croplands (9.58 kg m^−2^) > garden lands (9.07 kg m^−2^) > grasslands (8.07 kg m^−2^) > forestlands (7.35 kg m^−2^) > uncultivated lands (6.94 kg m^−2^). The SOC storage values in the 0.00–0.10 m, 0.00–0.20 m, 0.00–0.30 m and 0.00–1.00 m soil layers of Guizhou Province were 0.50, 0.87, 1.11 and 1.58 Pg, respectively. The SOC in the top 0.30 m of soil accounted for 70.25% of the total within the 0.00–1.00 m layer in Guizhou Province. It was concluded that assessing SOC storage in mountainous karst areas was more accurate when using land use rather than soil type. This result can supply a scientific reference for the accurate assessment of the SOC storage in the karst areas of southwestern China, the islands of Java, northern and central Vietnam, Indonesia, Kampot Province in Cambodia and in the general area of what used to be Yugoslavia, along with other karst areas with similar ecological backgrounds.

## 1. Introduction

Soil organic carbon (SOC) is a vital part of the terrestrial carbon pool and plays an important role in the global carbon cycle [1,2]. Previous studies have suggested that the global SOC stock is approximately 1550 Pg (petagrams of carbon or billion tons of carbon), which is far greater than the carbon stock stored in vegetation as plant components and in the atmosphere as carbon dioxide [3]. Therefore, a tiny variation in the carbon reserves of the SOC pool would cause a significant decrease or increase in the CO_2_ concentration of the atmosphere [4,5,6]. 

For more than two decades, SOC has attracted the attention of scientists globally, inspiring studies of the spatial distribution and storage of SOC at different scales. Therefore, it is of great importance to investigate and study the SOC storage capacity, profile distribution characteristics and the factors that impact SOC for the evaluation and management of carbon. This topic is especially important for developing and developed countries, because they are facing a trade-off between the need for economic development and the need to decrease carbon dioxide emissions.

Many studies concerning SOC stocks at different spatial scales have been conducted over the past several decades. Scharlemann et al. [7] found that the published values of the global SOC stock varied from 504 to 3000 Pg, with a mean value of 1460.5 Pg. In China, studies of the SOC stock at the national scale have been mainly based on data from the first and second national soil resource surveys. Based on the first national soil resource survey, Wang and Zhou [8,9] assessed that SOC storage in China was approximately 100.18 Pg. Pan [10] also calculated the SOC storage in China based on “Soil Species of China” and presented information from 2500 soil profiles, producing a result of 50 Pg of SOC storage. In another study, Wang et al. recalculated the SOC stock in China on the basis of the second national soil resource survey (2473 soil profiles) and found a value of 92.40 Pg of SOC storage [11]. However, when Wang et al. [12] adopted the soil taxonomy method and used two SOC densities to recalculate the SOC stock in China yet again (2473 soil profiles), the result was 61.5–121.1 Pg of storage. In a different study by Xie et al. [13], the SOC stock in China was estimated based on data from the second national soil resource survey, and different statistical methods were used. Their results showed that the SOC storage in China was approximately 50.6–154.0 Pg, with the mean storage found to be 102.30 ± 51.70 Pg. There have been many other studies concerning the SOC stock in China, and almost all of them are based on the first or second national soil resource survey, with the results ranging from 50 Pg to 154.0 Pg [14,15,16] of SOC storage.

There is a very large variance among the resulting SOC storage values. It is believed that there are many factors contributing to these differences, including the sampling methods (mainly referring to the richness of the original data sets) and statistical data analysis techniques. Increasingly, scientists have recognized that additional studies concerning the SOC at different spatial scales (national or regional) should be conducted to improve and perfect the global SOC database, and a great number of studies about the SOC stocks at the regional scale have been conducted over the past several years [17,18]. At present, many studies have been performed with the goal of estimating soil organic carbon storage at different spatial scales [19]. Large differences exist among the results at the same scales, even among the results that were gathered with the same method and in the same study region [20]. The main reasons for the differences in the estimation results are the differences in data sources, sample sizes, sampling depths and estimation methods. Insufficient numbers of samples and improper sample scales created inaccurate representations, resulting in inaccurate estimation results [21]. Few studies have focused on the SOC stock in Guizhou Province, Southwest China. Using the soil inventory database from the national soil resource survey and the historical collections from the Institute of Soil Sciences, Chinese Academy of Sciences, Li et al. [22] suggested that the total SOC in Guizhou Province was approximately 1.97 Pg in the 0.00–1.00 m soil layer. To our knowledge, this is the only study that refers specifically to the SOC stock in Guizhou Province. However, we think this value is controversial, since rock exposure, which is a key factor in the calculation of SOC in mountainous karst areas, was not taken into consideration in their calculation formulas.

It is well-recognized that mountainous karst areas exhibit high spatial heterogeneity in soil thickness, vegetation and soil nutrients, which presents an enormous challenge for the calculation of SOC stock in these regions [23]. Approximately 22.0 million km^2^ of the earth’s surface is composed of karst landforms [24]. However, published literature rarely examines the effects of rock exposure, which is an important factor that cannot be ignored in assessments of soil organic carbon. Some researchers mention the importance of rock exposure in assessments of SOC in karst regions, but they do not present a feasible strategy for addressing this problem [25]. In addition, the emissions from land use and land cover change (LULCC) are believed to be the second largest source of human-induced greenhouse gas emissions into the atmosphere (1.5 Pg/a) after fossil fuel combustion (5.3 Pg/a) (Intergovernmental Panel of Climate Change) [26,27]. 

In mountainous karst areas, the conflict between humans and the environment is a critical issue. Local people usually utilize soil resources based on landforms and intuitive soil characteristics, such as soil depth, soil fertility and rock exposure. Land use is an important factor controlling organic matter storage in soils, since it affects the amount and quality of litter input, litter decomposition rates and the processes of organic matter stabilization in soils [28]. Six et al. [29,30] suggested that organic matter is a major agent in stabilizing soil aggregates, and that soil organic matter (SOM) dynamics are related to the formation and destruction of slaking-resistant macro-aggregates. Therefore, it is advisable to calculate the SOC stock in mountainous karst areas based on land use [31]. The objective of this study was to calculate SOC storage in Guizhou Province based on different land uses, and to establish a feasible method for calculating SOC stock in mountainous karst areas. In this study, high-density sampling is used to accurately study the spatial distribution characteristics of soil organic carbon density, and an estimation method based on “land use type” organic carbon density is proposed, in order to provide scientific and technological support for the accurate estimation of soil organic carbon reserves in in Guizhou Province, and to be applicable to other mountainous karst regions globally.

## 2. Materials and Methods

### 2.1. Study Region

Guizhou Province is located on the eastern end of the Yunnan-Guizhou Plateau, which is one of the three main karst distribution regions of the earth. It covers an area of 176,155 km^2^ and has a population of 34.75 million people. Guizhou Province is a typical mountainous karst area, and there have been severe environmental impacts from human interference over the last eighty to ninety years of the past century [32]. Approximately 130,000 km^2^ land in Guizhou Province is composed of karst landforms, which is 73.7% of the total area (Table 1). This region is experiencing severe environmental problems, such as karst rocky desertification and persistent water shortages. The region has a subtropical humid monsoon climate. The mean annual temperature (MAT) and mean annual precipitation (MAP) range between 18 and 26 °C and between 1100 and 1300 mm of rain, respectively. The main ecosystem types include evergreen broad-leaved forest, coniferous and broad-leaved mixed forests, and montane elfin forest. Of the total land area in Guizhou Province, croplands account for 22.30%, garden lands for 0.58%, grasslands for 7.96%, forest lands for 39.06%, uncultivated lands for 24.00% (uncultivated land means land with no trees or shrubs, just several distributed wild grass species), and construction lands for 6.10% (here, construction land includes expressways, highways, tractor routes, railways, housing and the soil organic carbon (SOC) concentrations of these areas were not considered in the present study). Detailed information about land use for different districts in Guizhou Province is presented in Figure 1. 

### 2.2. Field Sampling

From March 2017 to September 2017, a total of 22,786 soil samples from 2854 soil profiles (from 0.00–1.00 m depths) were sampled in Guizhou Province in Southwest China (Figure 2a). A total of 1678 soil profiles were sampled from croplands, 66 soil profiles from garden lands, 142 soil profiles from grasslands, 563 soil profiles from forestlands, and 405 soil profiles from uncultivated lands (Figure 2b). Each profile was divided into 12 soil layers (0.00–0.05 m, 0.05–0.10 m, 0.10–0.15 m, 0.15–0.20 m, 0.20–0.30 m, 0.30–0.40 m, 0.40–0.50 m, 0.50–0.60 m, 0.60–0.70 m, 0.70–0.80 m, 0.80–0.90 m and 0.90–1.00 m) if the soil thickness was equal to or larger than 0.95 m [33]. Otherwise, the same sub-sampling scheme was utilized to the maximum depth of the sample. At the same time, the percent cover of the rock (rock exposure) around the sampling site was assessed with a tape measure. The soil bulk density (SBD) of each layer from all soil profiles was determined on the spot using the cylindrical core method. The slope and slope position of each sampling site were recorded simultaneously. The soil samples were stored in self-sealing plastic bags and returned to the laboratory where the soils were air dried, weighed and sieved to remove the gravel fraction (>2 mm grain size). The processed samples were saved in zip-lock bags for analysis of their SOC content. The total SOC content was determined by K_2_Cr_2_O_7_ oxidation at 170–180 °C followed by titration with 0.10 mol L^−1^ FeSO_4_ [34].

### 2.3. Calculation Methods and Statistical Analysis

By using field records and laboratory analysis data, we calculated the SOC density (SOCD) of each sampling site first, and the formula [22], which was improved with consideration of the rock outcrops, was utilized; this formula is as follows:
(1)$$SOCD=\overset{n+1}{\underset{m=1}{\sum }}C_{m}\times SBD_{m}\times T_{m}\times (1-\theta_{m})\times \left(1-A_{r}\right)$$
where *SOCD* is the SOC density (kg m^−2^) and *n* is the number of soil samples collected from the site. 

Because there was always a residual soil layer (<0.05 m) that could not be sampled as an independent layer, our calculation continued to the *n* + 1 layer, which was calculated by using the data of its upper layer and excluding the soil thickness. *C_m_*, *SBD_m_*, *T_m_*, and *θ_m_* represent the SOC concentration (g kg^−1^), soil bulk density (kg m^−3^), soil thickness (m) and gravel fraction (% of sample with grains larger than 2 mm) of layer *m*, respectively. *A_r_* is the percentage of rock cover (rock exposure) from the area surrounding the sampling site.

The storage of SOC (SSOC) in different land use types and the total SOC (0.00–1.00 m) in the study region were calculated via the following equations:
(2)$$SSOC_{i}=A_{i}\times (\bar{\sum_{l=1}^{k}SOCD_{l}})$$
(3)$$TSOC_{T}=\sum_{i=1}^{5}SSOC_{i}$$
where *SSOC_i_* is the storage of SOC (g) in land use *i*, *A_i_* is the coverage area (m^2^) of land use *i* in Guizhou Province, *k* is the number of total profiles sampled from land use *i*, and *TSOC_T_* is the total SOC storage (g) in the 0.00–1.00 m soil layer from the whole study region.

The calculations of SOC density and SOC storage were carried out in Microsoft Excel 2003 (Office 2003, Redmond, WA, USA). Analysis of Variance (ANOVA) analyses were performed using SPSS 13.0 (SPSS Inc., IBM Corporation, Chicago, IL, USA) and Origin 6.1 (Gamma Design Software, Origin 6.1, LLC Plainwell, MI, USA). Figures and maps were created with ArcMap 10.3 (ESRI, ArcMap 10.3, Redlands, CA, USA), (Figures and maps).

## 3. Results

### 3.1. Profile Characteristics of the SOC Concentrations for Different Land Uses

The mean SOC concentrations of the soil profiles (0.00–1.00 m) for each land use type shared a similar distribution pattern (Table 2). From 0.00 to 0.50 m, the mean SOC concentration decreased sharply and then fluctuated within a certain range (or decreased gradually) in the 0.50 to 1.00 m soil layer. The mean SOC concentrations (in the 0.00–1.00 m soil layer) ranged from 5.17 to 23.92 g kg^−1^ in the croplands; from 5.78 to 23.69 g kg^−1^ in the garden lands; from 6.37 to 29.87 g kg^−1^ in the grasslands; from 5.60 to 41.22 g kg^−1^ in the forestlands; and from 5.42 to 38.88 g kg^−1^ in the uncultivated lands. According to variance values, the SOC concentrations of all studied land uses were highly variable, but were especially variable for forestlands, uncultivated lands and grasslands. The SOC concentration profile (from 0.00–1.00 m deep) distribution models from croplands and garden lands were fitted to the quadratic equations *Y* = 0.0035*X*^2^ − 0.5229*X* + 24.789 (*R*^2^ = 0.4655) and *Y* = 0.0034*X*^2^ − 0.4881*X* + 23.617 (*R*^2^ = 0.3281), respectively. The SOC concentration profile distribution models from the forestlands, grasslands and uncultivated lands were fitted to the exponential equations *Y* = 38. 247*e*^−0.0281*x*^ (*R*^2^ = 0. 5328), *Y* = 24. 077*e*^−0.0199*x*^ (*R*^2^ = 0. 3776) and *Y* = 32. 245*e*^−0.0229*x*^ (*R*^2^ = 0. 3832), respectively. It was obvious that the studied land uses had mean SOC concentrations in the 0.00–0.30 m soil layer that decreased in the following order: Forestland > uncultivated land > grassland > cropland > garden land. Significant differences were detected between all land uses, with the exception of the difference between cropland and garden land. However, there was no significant difference between the SOC concentrations of the different land uses in the 0.60–1.00 m soil layer.

### 3.2. Effects of Land Use on Soil Bulk Density

Across all kinds of land uses studied here (Table 3), the mean SBD values showed similar trends with increasing soil depth. They increased in the top several soil layers, fluctuated in the several middle soil layers and finally decreased in the deeper soil layers. The mean SBD values of croplands increased from 1.20 × 10^3^ to 1.43 × 10^3^ kg m^-3^ in the top 0.30 m of soil, showed very little change in the 0.30 to 0.70 m layer and finally decreased from 1.39 to 1.34 × 10^3^ kg m^-3^ in the 0.70 to 1.00 m layer. The mean SBD values of the garden lands, grasslands and forestlands increased from 1.20 × 10^3^ to 1.39 × 10^3^ kg m^−3^, 1.19 × 10^3^ to 1.38 × 10^3^ kg m^-3^ and 1.10 × 10^3^ to 1.33 × 10^3^ kg m^−3^, respectively, in the top 0.00 to 0.60 m of soil. The SBD of these land uses decreased gradually at greater depths. The mean SBD values of the uncultivated lands increased from 1.14 × 10^3^ to 1.45 × 10^3^ kg m^-3^ in the 0.00 to 0.70 m soil layer, and then decreased slightly in the layer between 0.90 and 1.00 m. It is worth noting that the mean SBD value from the forestlands was the lowest among those for all land uses for the same soil layer. It is believed that the soil bulk density is closely associated with land use [35].

### 3.3. Profile Characteristics of the SOC Densities for Different Land Uses

The average SOC density of Guizhou Province was 8.96 kg m^−2^, which is slightly lower than the national SOC density of China (10.83 kg m^−2^ and 10.53 kg m^−2^) reported by Wang et al. [15,16]. As listed in Table 4, the average SOC density of the 0.00–0.10 m soil layer for the various land use areas decreased in the following order: Forestlands > uncultivated lands > croplands > grasslands > garden lands. However, the average SOC density of the whole profile (0.00–1.00 m) had a different order, as follows: Croplands > garden lands > grasslands > forest lands > uncultivated lands. The average soil thicknesses associated with the different land use areas decreased in the following order: Garden lands (0.75 m) > croplands (0.72 m) > grasslands (0.58 m) > uncultivated lands (0.42 m) > forestlands (0.42 m). The average rock exposure of the different land use areas increased in the following order: Croplands (8.94%) < garden lands (10.32%) < grasslands (22.04%) < forestlands (27.57%) < uncultivated lands (30.84%). It is obvious that soil thickness and rock exposure are both important factors affecting SOC density in mountainous karst regions. In addition, the results also indicate that soil thickness and rock exposure are crucial factors for local governments and farmers when making decisions about land use arrangements.

### 3.4. Storage of SOC in Guizhou Province

This study indicated that the top 1.00 m of soil in Guizhou Province stored approximately 1.58 Pg of SOC, and the 0.10 m, 0.20 m and 0.30 m deep soil layers of Guizhou Province stored 0.50 Pg, 0.87 Pg and 1.11 Pg of SOC, respectively. The total SOC storage in the top 1.00 m of soil of each district is presented in Figure 3. The SOC storage in the top 0.30 m of soil accounts for 70.25% of the total SOC storage within the top 1.00 m of soil in Guizhou Province. It is worth noting that the majority of the stored SOC was mainly distributed in the west, northwest, east and southeast parts of Guizhou Province. The different districts of Guizhou Province contained decreasing amounts of SOC storage (considering the soil from 0.00–1.00 m depth) in the following order: Bijie, Zunyi, Qiandongna, Qiannan, Tongren, Qianxinan, Liupanshui, Anshun and Guiyang; which held 0.43, 0.24, 0.23, 0.20, 0.14, 0.13, 0.08, 0.07 and 0.06 Pg of SOC, respectively. The total areas of Zunyi, Qiandongna, Bijie, Qiannan, Tongren, Qianxinan, Liupanshui, Anshun, and Guiyang were 30,780.73 km^2^, 30,278.06 km^2^, 26,847.52 km^2^, 26,191.78 km^2^, 18,006.41 km^2^, 16,785.93 km^2^, 9965.37 km^2^, 9253.06 km^2^ and 8046.67 km^2^, respectively. It is obvious that the land use types and geographic characteristics are critical factors affecting the SOC storage in this mountainous karst region. 

In the top 1.00 m of soil, the SOC storage in croplands, garden lands, grasslands, forestlands and uncultivated lands was 0.43 (0.4326) Pg, 0.01 (0.010) Pg, 0.13 (0.1301) Pg, 0.58 (0.5813) Pg and 0.42 (0.4228) Pg, respectively, and accounted for 27.42%, 0.68%, 8.25%, 36.85% and 26.93% of the total SOC storage, respectively. The SOC proportions of the different land uses in the different districts of Guizhou Province are shown in Figure 4.

## 4. Discussion

### 4.1. Factors Affecting the SOC Concentration

The soil bulk density is an important indicator of soil quality, since it reflects the soil’s ability to function as structural support, its water and solute movement and its aeration. Soil with high bulk density has worse soil porosity and soil compaction, leading to the restriction of root growth and poor movement of air and water through the soil. Compaction can result in shallow plant rooting and poor plant growth, influencing crop yield and reducing vegetative cover, both of which protect the soil from erosion [36]. Based on Table 2 and Table 4, it was found that negative relationships existed between the average SOC concentrations and the SBD values between different land uses in the top 40 cm soil layer. We believe that the basic reasons behind this phenomenon are the differences in land uses. The different land uses determine the inputs of particular types of plant biomass, such as litter and fine roots, resulting in significantly different aggregate contents and fractions of organic carbon. Liu et al. [28] found that land use strongly affected the amount of biomass input into the soil; the occurrence of aggregates, aggregate SOC and easily oxidizable organic carbon (EOC) from aggregates under grasslands and forestlands were normally higher than those under croplands. The present study found that the SOC contents in the 0.00–0.50 m soil layer in croplands and garden lands were much lower than those for grasslands, forestlands and uncultivated lands. Based on site investigation, the crop roots were mainly distributed in the 0.00 to 0.30 m layer, which may have been the main reason why the SOC contents were very low in the cropland and garden land soils in the 0.00–0.50 m layer in comparison with those of the other studied land uses. 

Crop roots absorb nutrients from soils efficiently, including some organic substances such as amino acids, proteins, starch and their degradation products. In addition, the distribution of the crop roots affects physical and chemical processes by changing the micro-circumstances within the soil.

Furthermore, crop roots provide a better setting for burrowing animals and microorganisms which benefits the biogeochemical degradation process of the organic matter in the soil. For the grasslands, the roots were distributed mainly in the 0.00–0.20 m soil layer. The grass roots were more abundant than the crop roots, but the SOC contents in the 0.00–0.30 m soil layer in the grasslands were higher than those of the croplands and garden lands. It is believed that three factors may contribute to this phenomenon. First, most crop biomass (above ground) will be removed by farmers after harvesting, and most grass biomass will remain and be incorporated into the soil after its death. Second, the grasslands had higher soil bulk densities, leading to worse soil porosity and soil compaction, consequently limiting the movement of air and water through the soil and restricting the bloom of soil animals and microorganisms. Therefore, the physical, chemical and biochemical degradation processes taking place underground were decreased in grasslands. Third, the root activities of most crops were higher than those of most grasses. Thus, organic matter in the soil would be more easily absorbed by root systems in the farmlands than in grasslands.

### 4.2. Factors Affecting SOC Density and Storage in Guizhou Province

Guizhou Province is a typical mountainous karst area, and karst landforms account for more than 70% of the total area. Therefore, soil depth and rock exposure are both crucial factors affecting the storage of SOC in this region. First, they are both factors in the formulas used to calculate SOC storage, and they directly influence the calculation results. Second, they are potential influential factors in decisions regarding land use. We believe that they directly and indirectly influence SOC sequestration in Guizhou Province.

The previous study resulted in a SOC stock in Guizhou Province of 1.97 Pg r [37]. That value is approximately 0.39 Pg larger than the result of our calculation. Their original data were based on an inventory database from the national soil resource survey and historical collections (only 2,456 soil profiles with 8714 representative diagnostic soil layers across China, which included Guizhou Province), while our data were based on field experiments and laboratory analyses (2,854 soil profiles comprised of 22,786 soil samples from Guizhou Province). In addition, they did not take rock exposure into consideration in their calculations, as we did in this study. Without the consideration of rock exposure, the SOC storage in the top 1.00 m of soil in Guizhou Province would be approximately 2.00 Pg based on our data, which is closer to the value reported by Li et al. As presented in Table 2, the average SOC concentrations of forestlands, uncultivated lands and grasslands were much higher than those of garden lands and croplands at the same depths. However, after considering soil thickness and rock exposure, the advantages for the average SOC density of the forest lands, uncultivated lands and grasslands were gradually eliminated as the soil depth increased.

The slope was also an important factor affecting SOC sequestration. As shown in Figure 5, rock exposure increased with increasing slope. However, the soil thickness decreased with increasing slope. On the basis of the 1,438 studied sites (with different slopes ranging from 1° to 84°), we found that the slope had a direct effect on the rock exposure, soil depth and SOC density, with Pearson coefficients of 0.238 (*p* < 0.01), −0.306 (*p* < 0.01) and −0.102 (*p* < 0.01), respectively. The present study also found that the slope position had no significant correlation with the SOC concentrations in the study region. However, soil thicknesses, rock outcrops and slopes were all important factors in the arrangement of land uses. Additionally, the SOC concentrations and SOC densities were closely associated with land use. Therefore, the assessment of SOC storage in the karst areas was more accurate considering land use than soil type.

The Guizhou Province had never before experienced as severe environmental impacts from human interference as those seen over the last eighty to ninety years of the past century [38]. To create more area for agricultural production, substantial amounts of forestlands and grasslands were reclaimed for croplands. However, along with the development of industry and society, the rural economic structure has improved dramatically in China [39]. A great number of young village labourers leave villages and are employed by various factories or companies in developed coastal cities, such as Hangzhou, Wenzhou, Ningbo, Shanghai, Xiamen and Guangzhou. The consequence is that only elderly people (many who are older than 50) remain for farming activities. As a result, many croplands are abandoned and become idle for several years. This phenomenon is very severe in the Bijie district in Guizhou Province. Since 2010, the provincial local governments have been trying to improve the ecological environment and to develop the tourism industry. One of the key actions has been to restore the forest system, and a great number of croplands have been reclaimed for forestation. In the present study, we instead used the SOC concentrations of the uncultivated lands with the mean values of the forestlands to assess the effects of this action on the SOC storage in Guizhou Province. The results indicated that approximately 2.22 × 10^11^ g carbon would be sequestrated in Guizhou Province as soil organic carbon in the future if this action is carried out effectively. Certainly, this value is just a conservative assessment, since a large amount of carbon will be sequestrated by the plants in the restored forest.

Karst landforms in Guizhou Province cover approximately 0.13 million km^2^, and the calculation error of SOC storage caused by the rock exposure was approximately 0.42 Pg. There are approximately 1.20 million km^2^ and 22.00 million km^2^ of karst landforms in China and in the world, respectively [40]. Based on our study, the calculation error caused by the rock exposure would reach 3.88 Pg and 71.08 Pg for China and the world, respectively, if the rock exposure factor was not taken into consideration in the calculation methods.

## 5. Conclusions

Based on the statistical analysis of the data from 2854 soil profiles, the SOC storage in Guizhou Province is estimated at 1.58 Pg in the top 1.00 m soil layer and at 1.11 Pg in the 0.00–0.30 m soil layer. The SOC storage in the 0.00–0.30 m soil layer accounted for 70.25% of the SOC storage in the top 1.00 m of soil in Guizhou Province. Land use, rock exposure, soil thickness and slope were key factors influencing the carbon sequestration in the soil in Guizhou Province. The present study strongly recommends that rock exposure be taken into consideration when assessing SOC storage in mountainous karst areas, since it directly influences the assessment results.

The present study suggests that the SOC calculation errors caused by rock exposure in China and around the world are approximately 3.88 Pg and 71.08 Pg, respectively. However, further studies are needed to correct these values, since many other factors also affect the SOC concentration (density/storage), such as climatic conditions, latitude, longitude and the intensity of human disturbance.

## Figures and Tables

**Figure 1 ijerph-16-04199-f001:**
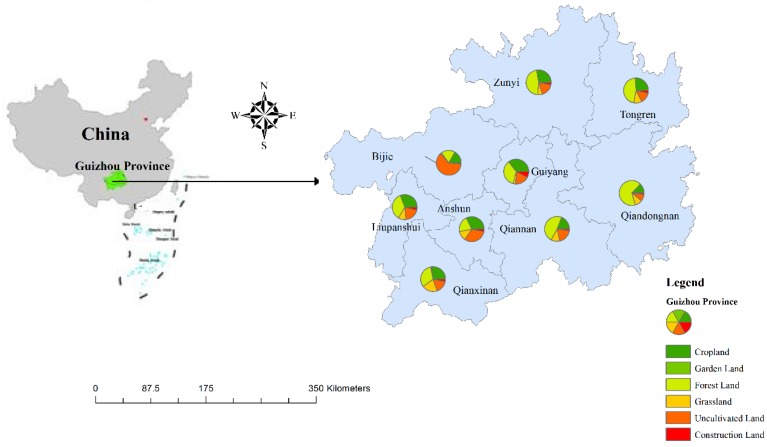
Land use in Guizhou Province, SW China.

**Figure 2 ijerph-16-04199-f002:**
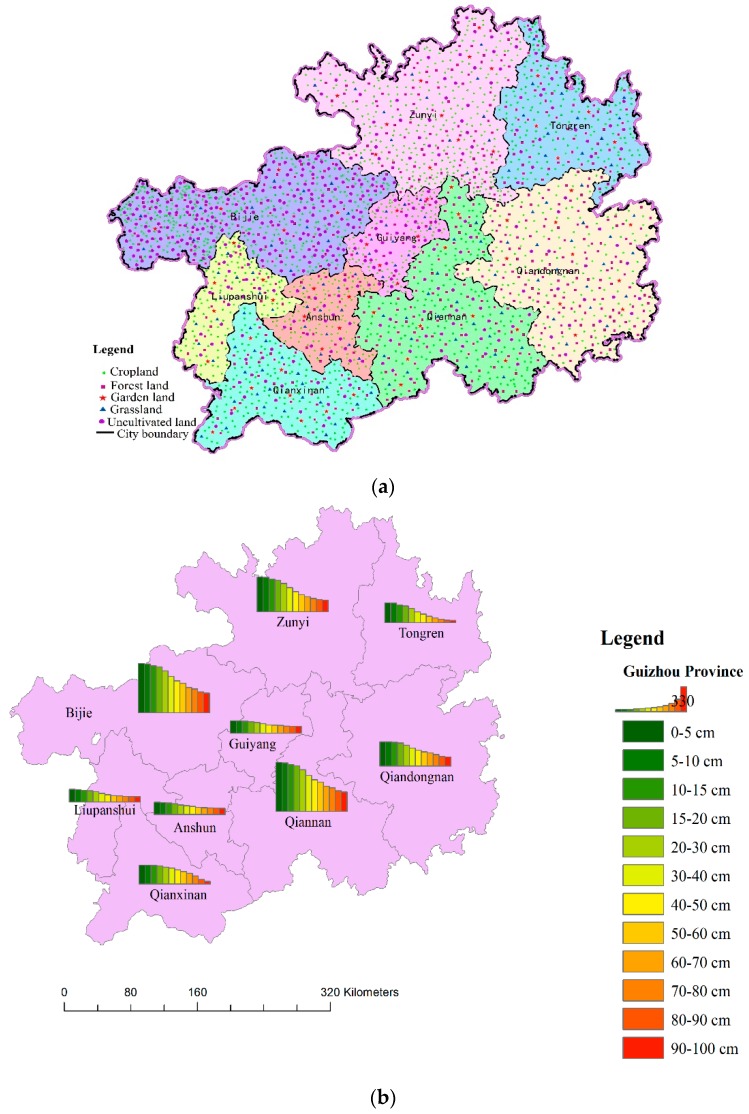
Distribution of sampling points and samples. (**a**) Distribution of sample points in different regions of Guizhou Province; (**b**) Distribution of sample points under different soil thickness.

**Figure 3 ijerph-16-04199-f003:**
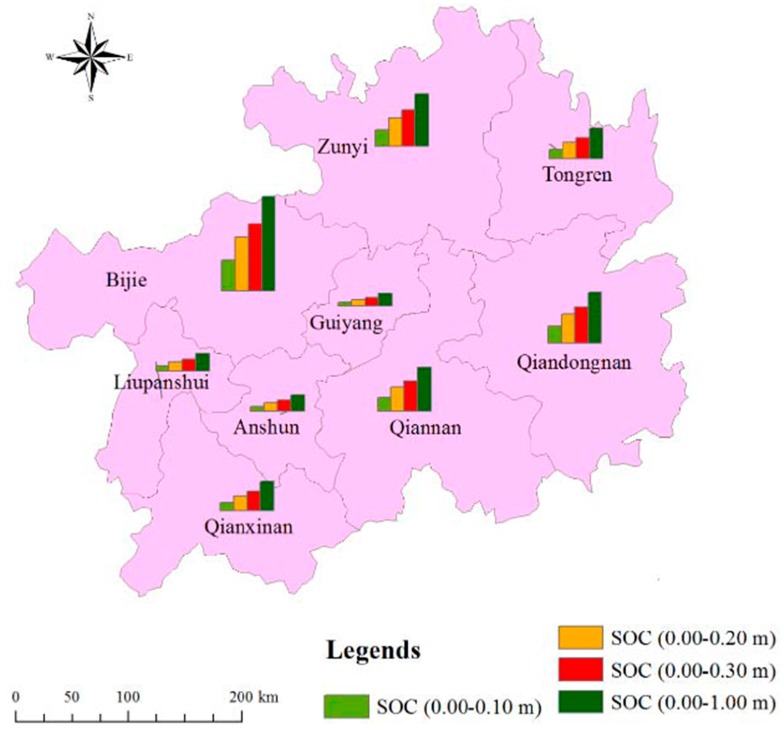
SOC storage in different soil layers in the different districts in Guizhou Province.

**Figure 4 ijerph-16-04199-f004:**
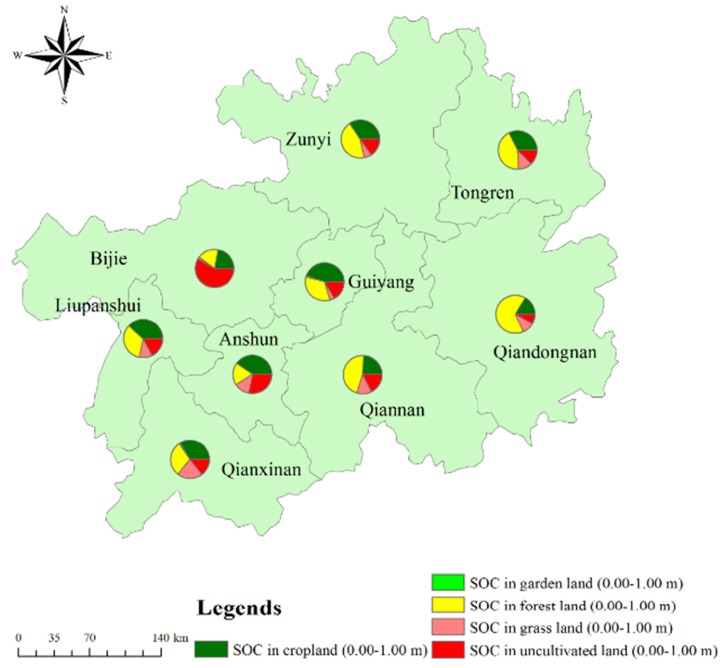
Proportions of SOC from different land uses in different districts in Guizhou Province.

**Figure 5 ijerph-16-04199-f005:**
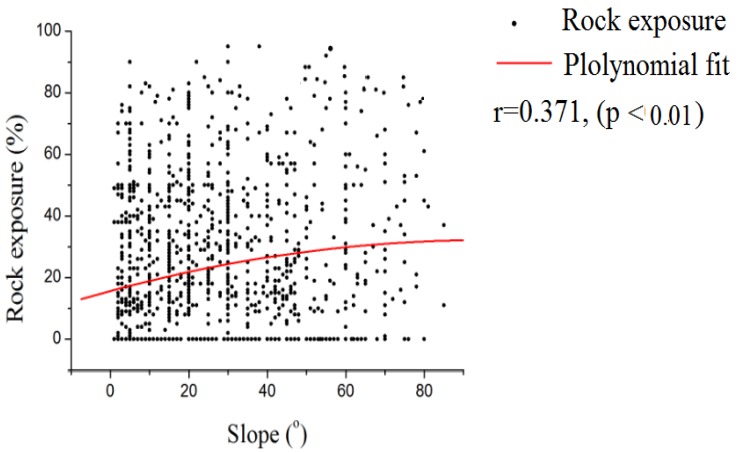
The relationship between slope and rock exposure

**Table 1 ijerph-16-04199-t001:** Distribution of rocky desertification within the study area

Districts	Unit	NRSD	LRSD	RDLD	RDMD	RDSD	RDESD	Non-Karst Region
Guiyang	area (km^2^)	2761.11	2192.05	1307.25	487.54	80.01	2.29	1203.64
	Percentage (%)	34.37	27.29	16.27	6.07	1.00	0.03	14.98
Qiannan	area (km^2^)	662.08	7015.02	4576.18	2098.91	847.92	194.48	4838.81
	Percentage (%)	3.27	34.67	22.62	10.37	4.19	0.96	23.91
Anshun	area (km^2^)	2475.85	1182.27	1243.53	954.51	601.06	170.61	2639.57
	Percentage (%)	26.72	12.76	13.42	10.30	6.49	1.84	28.48
Qianxinan	area (km^2^)	2985.97	2114.12	2111.54	1757.17	870.56	290.15	6674.49
	Percentage (%)	17.77	12.58	12.57	10.46	5.18	1.73	39.72
Liupanshui	area (km^2^)	1468.00	1529.49	1574.20	921.71	635.18	134.51	3650.71
	Percentage (%)	14.81	15.43	15.88	9.30	6.41	1.36	36.82
Bijie	area (km^2^)	6086.72	6597.99	4577.18	2066.50	312.94	51.76	7160.01
	Percentage (%)	22.67	24.57	17.05	7.70	1.17	0.19	26.66
Zunyi	area (km^2^)	8452.08	7109.84	3222.53	1275.83	173.81	1.59	10526.05
	Percentage (%)	27.48	23.11	10.48	4.15	0.57	0.01	34.22
Qiandongnan	area (km^2^)	2906.35	2345.66	1253.23	493.43	36.94	0.00	23301.39
	Percentage (%)	9.58	7.73	4.13	1.63	0.12	0.00	76.81
Tongren	area (km^2^)	3701.88	3940.14	2290.12	813.35	156.99	11.85	7088.67
	Percentage (%)	20.56	21.89	12.72	4.52	0.87	0.07	39.37

Note: NRSD = area with no risk of rocky desertification; LRSD = area with latent risk of rocky desertification; RDLD = area with a minor degree of rocky desertification; RDMD = area with a moderate degree of rocky desertification; RDSD = area with a severe degree of rocky desertification; RDESD = area with an extremely severe degree of rocky desertification.

**Table 2 ijerph-16-04199-t002:** Profile characteristics of the soil organic carbon (SOC) concentrations for different land uses (g kg^−1^).

Soil Layers	Cropland	Garden Land	Grassland	Forest Land	Uncultivated Land
0.00–0.05 m	23.92 ± 0.22Ja N = 1678	23.69 ± 1.32Ea N = 66	29.91 ± 1.28Gb N = 142	41.22 ± 0.86Hd N = 563	38.88 ± 1.00Hc N = 405
0.05–0.10 m	21.47 ± 0.20Ia N = 1678	20.49 ± 1.25Da N = 66	24.93 ± 1.08Fb N = 142	34.89 ± 0.75Gd N = 563	32.44 ± 0.89Gc N = 405
0.10–0.15 m	19.01 ± 0.20Ha N = 1674	17.47 ± 1.31Ca N = 64	21.47 ± 1.03Eb N = 142	30.07 ± 0.75Fd N = 556	27.35 ± 0.78Fc N = 395
0.15–0.20 m	16.01 ± 0.20Ga N = 1662	15.23 ± 1.24Ca N = 63	18.79 ± 1.00Db N = 118	25.57 ± 0.73Ed N = 526	23.76 ± 0.73Ec N = 343
0.20–0.30 m	12.51 ± 0.19Fa N = 1592	12.18 ± 1.10Ba N = 61	16.04 ± 0.99Cb N = 91	19.69 ± 0.62Dc N = 488	19.64 ± 0.74Dc N = 287
0.30–0.40 m	9.69 ± 0.16Ea N = 1552	9.55 ± 0.82Aba N = 52	12.76 ± 0.88Bb N = 64	14.30 ± 0.58Cbc N = 400	15.57 ± 0.75Cc N = 231
0.40–0.50 m	8.25 ± 0.15Da N = 1381	8.08 ± 0.64Aa N = 31	9.76 ± 0.65aABb N = 34	10.99 ± 0.61Bb N = 336	12.36 ± 0.74Bc N = 184
0.50–0.60 m	7.22 ± 0.15Ca N = 1209	7.77 ± 0.65Aab N = 27	8.45 ± 0.67Aab N = 19	9.21 ± 0.64Abb N = 267	9.01 ± 0.54Abb N = 156
0.60–0.70 m	6.68 ± 0.15Ca N = 1081	6.96 ± 0.73Aab N = 16	8.09 ± 0.70Aab N = 14	7.20 ± 0.58Aab N = 206	7.56 ± 0.51Ab N = 143
0.70–0.80 m	6.03 ± 0.15Ba N = 983	8.03 ± 1.58Aab N = 13	6.67 ± 0.70Aab N = 14	6.60 ± 0.53Aab N = 192	7.09 ± 0.53Ab N = 132
0.80–0.90 m	5.54 ± 0.15Aba N = 871	6.00 ± 0.58Aa N = 12	6.76 ± 0.79Aa N = 13	6.34 ± 0.53Aa N = 185	5.96 ± 0.51Aa N = 120
0.90–1.00 m	5.17 ± 0.15Aa N = 603	5.78 ± 0.61Aa N = 11	6.37 ± 0.78Aa N = 12	5.60 ± 0.46Aa N = 180	5.42 ± 0.42Aa N = 115

Means and standard errors. Within rows, the values followed by the same lowercase letter (a–d) are not significantly different (*p* < 0.05) for the same soil layer among different land uses; within the columns, the values followed by the same capital letter (A–J) are not significantly different (*p* < 0.05) among the soil layers of the same land use type; the significance was determined by Analysis of Variance (ANOVA).

**Table 3 ijerph-16-04199-t003:** Profile characteristics of soil bulk density from different land uses (×10^3^ kg m^−3^)

Soil Layers	Cropland	Garden Land	Grassland	Forestland	Uncultivated Land
0.00–0.05 m	1.20 ± 0.01Ac	1.20 ± 0.02Ac	1.19 ± 0.02Ac	1.10 ± 0.01Aa	1.14 ± 0.01Ab
0.05–0.10 m	1.25 ± 0.01Bc	1.26 ± 0.02Ac	1.24 ± 0.01Bbc	1.14 ± 0.01Ba	1.21 ± 0.01Bb
0.10–0.15 m	1.33 ± 0.01Cc	1.33 ± 0.03Bc	1.33 ± 0.02Cc	1.17 ± 0.01Ca	1.25 ± 0.01Cb
0.15–0.20 m	1.40 ± 0.01Ec	1.36 ± 0.02BCc	1.33 ± 0.02Cbc	1.20 ± 0.01Da	1.29 ± 0.01Db
0.20–0.30 m	1.43 ± 0.01Fc	1.35 ± 0.03BCb	1.35 ± 0.02Cb	1.24 ± 0.01DEa	1.33 ± 0.01Db
0.30–0.40 m	1.41 ± 0.01EFb	1.35 ± 0.03BCb	1.38 ± 0.02Cb	1.29 ± 0.01Ea	1.38 ± 0.02Eb
0.40–0.50 m	1.41 ± 0.01EFb	1.36 ± 0.03BCab	1.35 ± 0.02Cab	1.31 ± 0.01Ea	1.38 ± 0.02EFb
0.50–0.60 m	1.42 ± 0.02Fb	1.39 ± 0.03BCab	1.38 ± 0.02Cab	1.33 ± 0.02Ea	1.42 ± 0.02Fab
0.60–0.70 m	1.39 ± 0.01DEb	1.38 ± 0.03BCab	1.35 ± 0.03Cab	1.32 ± 0.02Ea	1.45 ± 0.02Fc
0.70–0.80 m	1.37 ± 0.01Db	1.42 ± 0.04Cbc	1.36 ± 0.03Cab	1.30 ± 0.02Ea	1.45 ± 0.03Fc
0.80–0.90 m	1.36 ± 0.01Db	1.46 ± 0.05Cc	1.31 ± 0.05BCab	1.30 ± 0.02Ea	1.46 ± 0.04Fc
0.90–1.00 m	1.34 ± 0.01CDb	1.38 ± 0.03 BCb	1.32 ± 0.03BCab	1.28 ± 0.02Ea	1.38 ± 0.03DEb

Means and standard errors. Within a row, the values followed by the same lowercase letter (a–d) are not significantly different (*p* < 0.05) between different land uses in the same soil layer; within columns, the values followed by the same capital letter (A–F) are not significantly different (*p* < 0.05) between soil layers of the same land use.

**Table 4 ijerph-16-04199-t004:** The average SOC densities of different land use areas in Guizhou Province (kg m^−2^)

Depth	Cropland	Garden land	Grassland	Forest Land	Uncultivated Land
0.00–0.10 m	2.32 ± 0.02AB	2.06 ± 0.09A	2.17 ± 0.09A	2.67 ± 0.06C	2.41 ± 0.07B
0.00–0.20 m	4.27 ± 0.04B	3.71 ± 0.18A	3.85 ± 0.17A	4.52 ± 0.11C	4.17 ± 0.12AB
0.00–0.30 m	5.63 ± 0.06B	4.87 ± 0.24A	5.08 ± 0.23A	5.62 ± 0.14B	5.25 ± 0.17A
0.00–0.40 m	6.61 ± 0.07B	5.81 ± 0.30A	5.95 ± 0.26A	6.23 ± 0.16A	5.94 ± 0.20A
0.00–0.50 m	7.37 ± 0.09B	6.59 ± 0.35A	6.54 ± 0.30A	6.61 ± 0.17AB	6.34 ± 0.22A
0.00–0.60 m	7.99 ± 0.10B	7.32 ± 0.42AB	6.98 ± 0.33A	6.85 ± 0.18A	6.58 ± 0.23A
0.00–0.70 m	8.51 ± 0.10B	7.91 ± 0.48B	7.34 ± 0.36AB	7.01 ± 0.19AB	6.73 ± 0.24A
0.00–0.80 m	8.94 ± 0.11B	8.45 ± 0.57B	7.63 ± 0.38AB	7.14 ± 0.19A	6.83 ± 0.24A
0.00–0.90 m	9.29 ± 0.12C	8.79 ± 0.61B	7.86 ± 0.40B	7.25 ± 0.20AB	6.89 ± 0.24A
0.00–1.00 m	9.58 ± 0.13C	9.07 ± 0.64BC	8.07 ± 0.42B	7.35 ± 0.20AB	6.94 ± 0.24A

Means and standard errors. Within rows, the values followed by the same capital letter (A–C) are not significantly different (*p* < 0.05) between the same soil layer for different land uses.

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
