# Peer review of "Discrepancies in Karst Soil Organic Carbon in Southwest China for Different Land Use Patterns: A Case Study of Guizhou Province"

_ijerph, 2019, doi:10.3390/ijerph16214199_

Round 1

Reviewer 1 Report

This manuscript describes a project to better characterize soil organic carbon (SOC) in Guizhou Province.  It would need a lot of work to become publishable, so I recommend that it be rejected in its current form.

The title itself is misleading.  The focus of the study is on SOC, although the title suggests it is about spatial heterogeneity in a karst landscape.  While the study does take place in a karst landscape, there is no assessment or incorporation of spatial heterogeneity into the study.  Moreover, the phrasing of the title suggests that land use patterns impact the spatial heterogeneity of the soil, which is not assessed. 

Your conclusion in the abstract is merely “that assessing SOC storage in karst mountain area is feasible.”  On one hand, that is a banal conclusion—we did the study, thus proving that we can do this study—without justifying much about why this is a notable conclusion.  On the other hand, you do not conduct any kind of assessment of the accuracy of the results, so while you may have proven that numbers can be calculated about SOC storage in a karst environment, you have no proof that these numbers are correct, meaning you have not actually proven that it is feasible to produce high quality results.

The development of the methods section needs much more work and elaboration.  Around line 60, you write that there are many factors that lead to a variance of assessed SOC sticks in China, “including sampling methods…and data statistic analysis technique. [sic]”  Please elaborate why those factors can lead to variance, and what the impacts of that variation can be.  What sampling methods and analysis techniques were used previously?  What among them worked well and worked poorly?  Answering these can then help you justify your methods choice, once you describe in more detail what that choice is.

You write on lines 82-83 that “local people, usually, utilize the soil resources based on the landform and soil intuitionistic characteristics.”    What are these intuitionistic characteristics?  Do local people use the soil for agriculture, to protect the forest, etc.?

In lines 98-99, what does “it had ever experienced the most severe contradiction between human and environment during the eighty and ninety years of the last century” mean?  I’m guessing environmental degradation during the 1980s and 1990s, and that this is an English transliteration of Chinese text, but it doesn’t translate effectively, word for word.  Also, how is this relevant?  Did it, and the desertification and water shortages you refer to afterwards, alter the SOC content?

Similarly, the word you want for “covered with strong concretes” in line 111 is “impervious”.

In Table 1, please add a row for the overall province, and provide row-based percents.  The districts are of different sizes, so percents will be more meaningful than square kilometers alone.

Can the pie-chart map in Figure 1 be raster-based instead?  The pie charts aggregate the data to the district level, which obscures the spatial heterogeneity your title suggests you should be studying.

What is the spatial distribution of the samples and soil profiles you describe in Section 2.2?  There’s no way to know the extent to which you study or account for spatial heterogeneity of soils without knowing how clustered or evenly spread the sample sites are, the extent to which they are spread throughout all provinces, or anything that might be used to create a semivariogram which could actually quantify the spatial heterogeneity of the data.

Either cite where you got equation 1 from, or justify it using the literature, explaining why this is an improvement over previous methods of estimating SOCD.  Your discussion seems to assert that the (1-Ar) term is new, but you don’t explain this up front.  Also, is equation 1 used aggregating all samples in a district and using the district-wide average?  Or is it by sample site, which you don’t say how many are in each district?  It seems to be a simple aggregate, although this kind of estimation is what the field of geostatistics and kriging was developed to do.  Kriging can incorporate covariate data, and account for, and describe, spatial heterogeneity.  It seems, though, that you did not use this technique.  Why not?  Or provide a better justification for the technique you did use to extrapolate from sample sites to districts and the province as a whole, recognizing the premise of the title, that spatial heterogeneity inherent in the karst landscape could mean the SOC/land use relationship in one district is different from the relationship in another district.

In Tables 2-4, how are the significant differences by row and column assessed?  The use of different letters suggests to me some form of ANOVA or MANOVA, but you never explain this, and an explanation of how the letter system works should be described in the main text, not just a table caption.  Basically, you say in line 150 that “Statistic[al] analysis were [sic] carried out” but never say what analyses you do.  How am I, as a reviewer or a reader, to know if you conducted the analysis properly if you don’t say what you do and why you chose those analytical methods?

Additionally, you report standard errors in tables 2 and 3, acknowledging uncertainty in these results, yet when you provide the total SOC in lines 195 and 196 as 1.58 Pg, you don’t provide a standard error here.  The inputs to this calculation have uncertainty, so why not carry that out to the end of the analysis?  Without it, your 1.58 Pg conclusion seems to have a lot of uncertainty that you don’t acknowledge.  Likewise, later on, in line 319, you simply say that “the calculation error of SOC storage caused by rock exposure is about 0.42 Pg”.  This statement implies that the numbers used for this comparison are accurate, and that the difference between them of 0.42 Pg must therefore be attributable to rock exposure.

The lists in section 3.3, lines 200-205 need to be presented as a choropleth map of the SOC standardized by the area of the province.  This might make your next statements “obvious”, because the lists alone do not accomplish this.  Meanwhile, the map in Figure 2 appears to not be standardized by area, but without a vertical scale or legend for the height component of the bar charts, it is impossible to tell for certain.  Are the bars for Guiyang shorter than those for Bijie because Bijie is larger, or because Bijie indeed has more SOC per square kilometer?  Likewise, a comparison of Figures 1 and 3 might help, or find another means of mapping the same information than pie charts per district.  For example, I see that Qiangdongnan has more SOC in forest land than anywhere else.  Is that because of the spatial heterogeneity allowing Qiangdongnan’s forests to better capture SOC, or simply because Qiangdongnan has more forest?

I’m wondering why section 4.1 on land use versus soil bulk density is in the Discussion section instead of the Results section.  In this, on line 229, you vaguely assert “it is believed that the soil bulk density is closely associated [with] land use”.  Believed by whom?  Cite this.

In lines 241 and 242, you assert that “based on tables 2 and 3, it is found that negative relationships exist between average SOC concentrations and SBD values”.  I think you mean tables 2 and 4, and why not use a regression analysis for this?  Looking at two tables on different pages is difficult and does not readily reveal any relationships at all.

In lines 254 and 255, what are “production daughters” and “micro-circumstances”?

Please provide the R-squared values for the regressions in Figure 4.  They may well be statistically significant, by virtue of a large sample, but it doesn’t mean there’s much scientific significance to interpret with it, if R-squared is very low.  The variance of points around the trend lines in Figure 4 suggests to me that R-squared would, indeed, be very low.  This makes me reluctant to have much faith in the soundness of these results.

Author Response

Thanks a lot to the contribution by the editors and reviewers. All the comments are precious and indispensable for improvement of our manuscript.

Based on your comments and suggestions, we have revised the manuscript carefully. The replies to referees are listed one by one. The language has been revised by American Journal Experts (AJE) company, and also double checked by a native English speaker throughout. Furthermore, the authors have reorganized the manuscript and the logic of manuscript has been improved. In addition, all the amendments in this revised paper are marked in blue.

Reviewers 1#:

Comment 1: The title itself is misleading. The focus of the study is on SOC, although the title suggests it is about spatial heterogeneity in a karst landscape. While the study does take place in a karst landscape, there is no assessment or incorporation of spatial heterogeneity into the study.  Moreover, the phrasing of the title suggests that land use patterns impact the spatial heterogeneity of the soil, which is not assessed.

Reply: Thanks for the reviewer’s suggestion. This work was trying to reflect the karst soil spatial heterogeneity via soil organic carbon heterogeneity. According to the reviewer’s suggestion, the title has been modified as “Discrepancies in Karst Soil Organic Carbon in Southwest China under Different Land Use Patterns: A Case Study of Guizhou Province”.

Comment 2: Your conclusion in the abstract is merely “that assessing SOC storage in karst mountain area is feasible.” On one hand, that is a banal conclusion—we did the study, thus proving that we can do this study—without justifying much about why this is a notable conclusion.  On the other hand, you do not conduct any kind of assessment of the accuracy of the results, so while you may have proven that numbers can be calculated about SOC storage in a karst environment, you have no proof that these numbers are correct, meaning you have not actually proven that it is feasible to produce high quality results.

Reply: Thanks for the reviewer’s suggestion. The conclusion in the abstract has been rewritten yet. Based on present study, we believe that assessing SOC storage in karst mountainous area is more accurate via land use rather than via soil type. For one thing, thickness, rock outcrop and slope are impotent factors in soil quality which are potential effect local people in arrangement of land use. For another thing, great discrepancies of SOC concentration existed among different land uses. Therefore, assessing SOC storage in karst mountainous area via land use not only consider the environmental factor, the subjective effects of local people were also included.

Comment 3: The development of the methods section needs much more work and elaboration.  Around line 60, you write that there are many factors that lead to a variance of assessed SOC sticks in China, “including sampling methods…and data statistical analysis technique. [sic]” Please elaborate why those factors can lead to variance, and what the impacts of that variation can be. 

Reply: Thanks for the reviewer’s suggestion. Many scholars have carried out a lot of studies on estimating soil organic carbon storage at different scales. Great differences existed among the results at the same scale, even among the results via the same method and the same study region. The difference of data source, sample size, sampling depth and estimation method is the main reason for the difference of estimation results. The insufficient number of samples and improper sample scale make the representation inaccurate, resulting in inaccurate estimation results. At present, assessment of SOC storage was mainly calculated from soil organic matter content and bulk density (Refer. 19).

Comment 4: What sampling methods and analysis techniques were used previously?  What among them worked well and worked poorly? 

Reply: Thanks for the reviewer’s suggestion. This method was not well suitable for assessment of SOC storage in karst mountainous area. Firstly, soil organic matter content and bulk density are of high heterogeneity rather than non-karst area; secondly, land use pattern is also much complex in comparison to non-karst area. Based on present study, land use is a critical factor affecting SOC concentration. In addition, soil properties and environmental factors are key potential factors in arrangement of land use. Therefore, assessment of SOC storage in karst mountainous area via land use would be more accurate rather than traditional method (just the product of SOC concentration and bulk density).

Comment 5: You write on lines 82-83 that “local people, usually, utilize the soil resources based on the landform and soil intuitionistic characteristics.” What are these intuitionistic characteristics?  Do local people use the soil for agriculture, to protect the forest, etc.?

Reply: Here, intuitionistic characteristics mainly include soil thickness, rock outcrop, slope and soil fertility. Lands with greater soil thickness, lower rock outcrop and smaller slope were arranged as croplands, and the other lands were left and become forest lands, grass lands or idle lands under the work of natural succession.

Comment 6: In lines 98-99, what does “it had ever experienced the most severe contradiction between human and environment during the eighty and ninety years of the last century” mean?  I’m guessing environmental degradation during the 1980s and 1990s, and that this is an English transliteration of Chinese text, but it doesn’t translate effectively, word for word.  Also, how is this relevant? Did it, and the desertification and water shortages you refer to afterwards, alter the SOC content?

Reply: Thanks for the reviewer’s suggestion. During the 1960s and 1970s, the high increase rate of China population caused a large population in the 1980s and 1990s. Consequently, the food production could not satisfy the people needs. Therefore, a large amount of forests, grass lands, etc. were destroyed and reclaimed as croplands. This event led to deterioration of ecological system (including water shortage and occurrence of rocky desertification).

Comment 7: The words you want for “covered with strong concretes” in line 111 is “impervious”.

Reply: Thanks! This sentence has been rephrased.

Comment 8: In Table 1, please add a row for the overall province, and provide row-based percents. The districts are of different sizes, so percents will be more meaningful than square kilometers alone.

Reply: Thanks! Row-based percents have been added.

Comment 9: Can the pie-chart map in Figure 1 be raster-based instead? The pie charts aggregate the data to the district level, which obscures the spatial heterogeneity your title suggests you should be studying.

Reply: Thanks for the reviewer’s suggestion. The Figure 1 has been revised to become clearer and more representative and the Figure 1 presents the status of land use in different districts of Guizhou Province. It is not advisable to change it into raster-based map. For one thing, this map presents information of six kinds land uses in different districts. If it was presents as raster-based map, six figures will be needed at least. For another thing, the shape of different districts is not regular, it is unreasonable to present this information in raster-based map.

Comment 10: What is the spatial distribution of the samples and soil profiles you describe in Section 2.2? There’s no way to know the extent to which you study or account for spatial heterogeneity of soils without knowing how clustered or evenly spread the sample sites are, the extent to which they are spread throughout all provinces, or anything that might be used to create a semi variogram which could actually quantify the spatial heterogeneity of the data.

Reply: Thanks for the reviewer’s suggestion. The sampling sites (soil profiles) were designed random from different land uses in different districts.

Comment 11: Either cite where you got equation 1 from, or justify it using the literature, explaining why this is an improvement over previous methods of estimating SOCD.  Your discussion seems to assert that the (1-Ar) term is new, but you don’t explain this up front.  Also, is equation 1 used aggregating all samples in a district and using the district-wide average?  Or is it by sample site, which you don’t say how many are in each district?  It seems to be a simple aggregate, although this kind of estimation is what the field of geostatistics and kriging was developed to do. Kriging can incorporate covariate data, and account for, and describe, spatial heterogeneity.  It seems, though, that you did not use this technique.  Why not?  Or provide a better justification for the technique you did use to extrapolate from sample sites to districts and the province as a whole, recognizing the premise of the title, that spatial heterogeneity inherent in the karst landscape could mean the SOC/land use relationship in one district is different from the relationship in another district.

Reply: Thanks! The equation 1 was mainly cited from the reference [19]. However, we have improved the original equation with consideration of rock outcrop, which is of great importance in assessment of SOC density or SOC storage in karst areas. Just as presented in the Figure 1, the distribution of land uses different from each district. We believe it is unreasonable to estimation SOC density or SOC storage via kriging technique.  

Comment 12: In Tables 2-4, how are the significant differences by row and column assessed?  The use of different letters suggests to me some form of ANOVA or MANOVA, but you never explain this, and an explanation of how the letter system works should be described in the main text, not just a table caption.  Basically, you say in line 150 that “Statistic[al] analysis were [sic] carried out” but never say what analyses you do.  How am I, as a reviewer or a reader, to know if you conducted the analysis properly if you don’t say what you do and why you chose those analytical methods?

Reply: Thanks! This is our negligence.The significant differences by row and column assessed according to ANOVA analysis. We have revised the sentence in line 150 (previous vision). Statistic analysis were carried out with Microsoft excel 2003 (caculations of SOC density and SOC storage), SPSS 13.0 for windows (ANOVA analysis), Origin 6.1 and Arc Map 10.3 (Figures and maps).

Comment 13: Additionally, you report standard errors in tables 2 and 3, acknowledging uncertainty in these results, yet when you provide the total SOC in lines 195 and 196 as 1.58 Pg, you don’t provide a standard error here.  The inputs to this calculation have uncertainty, so why not carry that out to the end of the analysis? Without it, your 1.58 Pg conclusion seems to have a lot of uncertainty that you don’t acknowledge. Likewise, later, in line 319, you simply say that “the calculation error of SOC storage caused by rock exposure is about 0.42 Pg”.  This statement implies that the numbers used for this comparison are accurate, and that the difference between them of 0.42 Pg must therefore be attributable to rock exposure.

Reply: Thanks! Values in tables 2 and 3 are mean values. Therefore, there were standard error values for them. There is only one value of the total SOC storage in top 1.00 m soil horizons of Guizhou Province (1.58 Pg).

Comment 14: The lists in section 3.3, lines 200-205 need to be presented as a choropleth map of the SOC standardized by the area of the province.  This might make your next statements “obvious”, because the lists alone do not accomplish this.  Meanwhile, the map in Figure 2 appears to not be standardized by area, but without a vertical scale or legend for the height component of the bar charts, it is impossible to tell for certain.  Are the bars for Guiyang shorter than those for Bijie because Bijie is larger, or because Bijie indeed has more SOC per square kilometer?  Likewise, a comparison of Figures 1 and 3 might help, or find another means of mapping the same information than pie charts per district.  For example, I see that Qiangdongnan has more SOC in forest land than anywhere else. Is that because of the spatial heterogeneity allowing Qiangdongnan’s forests to better capture SOC, or simply because Qiangdongnan has more forest?

Reply: Thanks for the reviewer’s suggestion. In Figure 2, hight of bars represent storage of SOC. There are some factors contribute to this discrepancy. We believe key factors including covered area and land use pattern.

Comment 15: I’m wondering why section 4.1 on land use versus soil bulk density is in the Discussion section instead of the Results section.  In this, on line 229, you vaguely assert “it is believed that the soil bulk density is closely associated [with] land use”.  Believed by whom?  Cite this.

Reply: Thanks for the reviewer’s suggestion. The key issue of this study was SOC. Therefore, soil bulk density was arranged as an affecting factor and presented in discussion section. Literature has be added on line 229 (previous vision) “it is believed that the soil bulk density is closely associated with land use”.

Comment 16: In lines 241 and 242, you assert that “based on tables 2 and 3, it is found that negative relationships exist between average SOC concentrations and SBD values”.  I think you mean tables 2 and 4, and why not use a regression analysis for this?  Looking at two tables on different pages is difficult and does not readily reveal any relationships at all.

Reply: Yes, that should base on tables 2 and 4. Thanks for reviewer’s suggestion. This negative relationship could be found easily by comparison of two tables. We have revised this sentence to make it more obvious.

Comment 17: In lines 254 and 255, what are “production daughters” and “micro-circumstances”?

Reply: Here, “production daughters” referring to their degradation products.  “micro-circumstances” means the circumstance surrounding the crop roots. Thanks!

Comment 18: Please provide the R-squared values for the regressions in Figure 4.  They may well be statistically significant, by virtue of a large sample, but it doesn’t mean there’s much scientific significance to interpret with it, if R-squared is very low.  The variance of points around the trend lines in Figure 4 suggests to me that R-squared would, indeed, be very low.  This makes me reluctant to have much faith in the soundness of these results.

Reply: Thanks for the reviewer’s suggestion. Figure 4 has been revised and added R-squared. The R-squared values for the regressions in Figure 4(a) and Figure 4(b) is 0.371 (p<0.001) and 0.458 (p<0.001), respectively. We have added this information in Figure 4. We thought this Figure would present readers a variation tendency of rock outcrop and soil thickness along with increase of slope gradient.

The other minor questions have been revised directly in the new edition of manuscript (marked in blue).

If you have any further question about this paper, please don’t hesitate to let me know.

Thank you and all the referees very much for the kind advice.

Sincerely yours,

Yunchao Zhou

Reviewer 2 Report

This is a very well written paper assessing the soil organic carbon (SOC) in a karst mountainous area in Southwest China. It is straightforward results and mainly descriptive. There are some grammar and english issues and I have attached my suggested revisions.

Author Response

Thanks a lot to the contribution by the editors and reviewers. All the comments are precious and indispensable for improvement of our manuscript.

Based on your comments and suggestions, we have revised the manuscript carefully. The replies to referees are listed one by one. In addition, all the amendments in this revised paper are marked in blue.

Reviewers 2#

This is a very well written paper assessing the soil organic carbon (SOC) in a karst mountainous area in Southwest China. It is straightforward results and mainly descriptive. There are some grammar and English issues and I have attached my suggested revisions.

Comment 1: There are some grammar and English issues.

Reply: Thanks for the constructive comments. The language has been revised by American Journal Experts (AJE) company, and also double checked by a native English speaker throughout. Furthermore, the authors have reorganized the manuscript and the logic of manuscript has been improved.

Comment 2:

Reply: The other suggested revisions and comments in the manuscript.

Reply: Thank you very much for your careful revision. The suggested revisions and comments have been revised.

The other minor questions have been revised directly in the new edition of manuscript (marked in blue).

If you have any further question about this paper, please don’t hesitate to let me know.

Thank you and all the referees very much for the kind advice.

Sincerely yours,

Yunchao Zhou

Reviewer 3 Report

a) The present content about research gaps is too general and I would suggest improve the statement about the research gaps. It would be helpful for readers if the contribution of your study to narrow the research gaps. b) What’s strategy for sample? c) For (b) in Figure 4, is the linear fit proper?

Author Response

Thanks a lot to the contribution by the editors and reviewers. All the comments are precious and indispensable for improvement of our manuscript.

Based on your comments and suggestions, we have revised the manuscript carefully. The replies to referees are listed one by one. The language has been revised by American Journal Experts (AJE) company, and also double checked by a native English speaker throughout. Furthermore, the authors have reorganized the manuscript and the logic of manuscript has been improved. In addition, all the amendments in this revised paper are marked in blue.

Reviewers 3#

Comment 1: The present content about research gaps is too general and I would suggest improve the statement about the research gaps. It would be helpful for readers if the contribution of your study to narrow the research gaps.

Reply: This study focusses on SOC in karst area. As present in introduction section, “karst mountainous areas are of high spatial heterogeneity in soil thickness, vegetation and soil nutrients.”. It is a huge challenge to calculate SOC stock accurately in these regions. We are just on the way to make it more accurate.

Comment 2: What’s strategy for sample?

Reply: To avoid the subjective, the sampling sites were designed randomly in different land use in different districts.

Comment 3: For (b) in Figure 4, is the linear fit proper?

Reply: Figure 4(b) presents the relationship between slope and soil thickness. We have try some methods to fit this relationship, and the linear fit is best one (the R-squared value is greatest).

The other minor questions have been revised directly in the new edition of manuscript (marked in blue).

If you have any further question about this paper, please don’t hesitate to let me know.

Thank you and all the referees very much for the kind advice.

Sincerely yours,

Yunchao Zhou

Round 2

Reviewer 3 Report

 I have no further comments for authors.

Author Response

Thanks a lot to the contribution by the editors and reviewers. All the comments are precious and indispensable for improvement of our manuscript.

Based on your comments and suggestions, we have revised the manuscript carefully. The replies to referees are listed one by one. The language has been revised by American Journal Experts (AJE) company, and also double checked by a native English speaker throughout. Furthermore, the authors have reorganized the manuscript and the logic of manuscript has been improved. In addition, all the amendments in this revised paper are marked in blue.

Comment 1:Introduction, (Lines 69-74, page 2) – Please check the English style and provide references for this sentence.

Reply: Thanks for the reviewer’s constructive comments. The introduction has been revised by American Journal Experts (AJE) company, and also double checked by a native English speaker throughout. More references have been added in the Introduction (see Lines 69-74, page 2).

Comment 2: (Lines 83-85, page 2) – Please provide references 
Reply: According to the comments, we have added some references in the introduction (Lines 83-85, page 2). Thanks!

Comment 3:Material & Methods ,Lines 121, page 3 - remove "and so on" from your sentence.
Reply: According to the comments, we have removed "and so on" from our manuscript (Lines 121, page 3). Thanks!

Comment 4:Figure 1. It is unnecessary to show the world map 
Reply: Thanks for the reviewer’s criticism. The Figure 1 has been removed the world map .

Comment 5:Subsection 2.2 Field Sampling - For field samples, provide the number of observations per soil layer and scattered by district. In addition, the time period under analysis and the corresponding seasonal time when you collected the 22786 soil samples. You can include this information in table 2.
Reply: Thanks for the reviewer’s constructive comments. We have added the area distribution of sampling points and the number of sections in Figure 2. The number of sample points has been supplemented in Table 2, and other sampling information has been added in Subsection 2.2.

Comment 6: (Lines 132 - 137, page 4) - Several details are difficult to understand and are missing in this paragraph. To improve readability, I suggest that you include descriptive information in a table or provide a diagram.
Reply: Thanks for the reviewer’s constructive comments. We have added the area distribution of sampling points and the number of sections in Figure 2.

Comment 7:The authors replied that “the sampling sites were randomly designed on different land uses in different districts” [Reviewer#1, 10 comment and Reviewer# 3 , comment 2]. The authors can provide the methodology applied to the randomly selected samples (soil profile) (references?). Indeed, to be more precise, provide a map that identifies the centroids of the soil samples just to see if they are scattered across the different Guizhou province / district. In addition, information on soil type in each district / sampling site will be appreciated.
Reply: Thanks for the reviewer’s constructive comments. We have added a figure of the centroids of the soil samples, and we have added a reference in the introduction (Lines 133-136, page 4).

Comment 8:Discussion ,Sub-section 4.1. Effects of land use on soil bulk density.  I agree with Reviewer#1 (comment 15 , Report 1) that the Soil Bulk density (SBD) should be moved to the Results section. It is very unusual to see results as Table 4 in a Discussion section. Moreover, the Soil Bulk Density is half of the results as expressed in equation 1, to calculate the Soil Organic Carbon Density (SOCD). The authors included an explanation of how they assessed the SBD in 2.2. Field Sampling. So definitely, Table 4 should appear in the Results section. 
Reply: Thanks for the reviewer’s constructive comments. We have moved Sub-section 4.1 to the Results section.

Comment 9:4.2 . Affecting factors for the SOC concentration – Please, change for readability to “ Factors affecting SOC concentration” 
Reply: Thanks for the reviewer’s constructive comments. I have changed the title of section 4.2

Comment 10:4.3. Affecting factors for SOC density and storage in Guizhou Province – Please, change to readability “ Factors affecting SOC density and storage in Guizhou Province” 

Reply: Thanks for the reviewer’s constructive comments. I have changed the title of section 4.3

Comment 11:a) The present content about research gaps is too general and I would suggest improve the statement about the research gaps. It would be helpful for readers if the contribution of your study to narrow the research gaps. b) What’s strategy for sample? c) For (b) in Figure 4, is the linear fit proper?

Reply: Thanks for the reviewer’s constructive comments. a)  In this study, high-density sampling is used to accurately study the spatial distribution characteristics of soil organic carbon density, and an estimation method based on "land use type" organic carbon density is proposed, in order to provide scientific and technological support for the accurate estimation of soil organic carbon reserves in plateau karst small watershed. The end of introduction has been revised, and objectives of the research are more clear. b) We have added a figure of the centroids of the soil samples, and we have added a reference in the introduction. b) According to the comments, we have removed For (b) in Figure 4, and the research contents of soil thickness and slope are described in the manuscript.

The other minor questions have been revised directly in the new edition of manuscript.

If you have any further question about this paper, please don’t hesitate to let me know.

Thank you and all the referees very much for the kind advice.

Sincerely yours,

Yunchao Zhou
